# Prospects of Electrocorticography in Neuropharmacological Studies in Small Laboratory Animals

**DOI:** 10.3390/brainsci14080772

**Published:** 2024-07-31

**Authors:** Yuriy I. Sysoev, Sergey V. Okovityi

**Affiliations:** 1Pavlov Institute of Physiology, Russian Academy of Sciences (RAS), Saint Petersburg 199034, Russia; 2Department of Neuroscience, Sirius University of Science and Technology, Sirius Federal Territory 354340, Russia; 3Institute of Translational Biomedicine, Saint Petersburg State University, Saint Petersburg 199034, Russia; 4Department of Pharmacology and Clinical Pharmacology, Saint Petersburg State Chemical Pharmaceutical University, Saint Petersburg 197022, Russia; okovityy@mail.ru; 5N.P. Bechtereva Institute of the Human Brain, Saint Petersburg 197022, Russia

**Keywords:** electroencephalography, electrocorticography, ECoG, neuropharmacology, pharmacoencephalography

## Abstract

Electrophysiological methods of research are widely used in neurobiology. To assess the bioelectrical activity of the brain in small laboratory animals, electrocorticography (ECoG) is most often used, which allows the recording of signals directly from the cerebral cortex. To date, a number of methodological approaches to the manufacture and implantation of ECoG electrodes have been proposed, the complexity of which is determined by experimental tasks and logistical capabilities. Existing methods for analyzing bioelectrical signals are used to assess the functional state of the nervous system in test animals, as well as to identify correlates of pathological changes or pharmacological effects. The review presents current areas of applications of ECoG in neuropharmacological studies in small laboratory animals. Traditionally, this method is actively used to study the antiepileptic activity of new molecules. However, the possibility of using ECoG to assess the neuroprotective activity of drugs in models of traumatic, vascular, metabolic, or neurodegenerative CNS damage remains clearly underestimated. Despite the fact that ECoG has a number of disadvantages and methodological difficulties, the recorded data can be a useful addition to traditional molecular and behavioral research methods. An analysis of the works in recent years indicates a growing interest in the method as a tool for assessing the pharmacological activity of psychoactive drugs, especially in combination with classification and prediction algorithms.

## 1. Introduction

Neurophysiological methods make it possible to solve a wide range of experimental problems in neurobiology. For example, functional magnetic resonance imaging (fMRI) and positron emission tomography (PET) due to high spatial resolution may provide valuable information on the involvement of brain areas in the response to various challenges. The direct recording of activity of the cerebral cortex (electrocorticography, ECoG) can be used to assess its functional state, which is potentially applicable in modeling various pathological conditions and searching for new therapeutic approaches to their correction. In comparison with neuroimaging methods based on the assessment of hemodynamic or neurochemical processes, such as fMRI and PET, ECoG signal has a high temporal resolution, which allows for the study of neuronal activity in ranges of tens of milliseconds. Thus, a large amount of useful information about the brain bioelectrical activity can be obtained using Fourier transform and signal decomposition into frequencies. Studying the responses of the cerebral cortex to various stimuli (e.g., visual or somatosensory) provides information about the operation of sensory systems. The determination of ECoG correlates of activity and emotional aspects of animal behavior is used to study the physiology of behavior or the activity of new psychoactive drugs.

Despite the many advantages of EEG and ECoG in neurobiological studies, only a small number of researchers use them in their work, giving preference to behavioral tests, supplementing them with molecular genetic, biochemical, and histological methods. All of the above have proven themselves in neuropharmacological studies, but each approach has its own drawbacks. For example, basic behavioral tests do not always clearly identify the pharmacological effect of a particular drug. For instance, a potentially antipsychotic agent can decrease exploratory activity in the open field test or elevated plus maze, and, therefore, may be misclassified as a sedative agent, but “true” antipsychotic activity can be verified only by specialized tests such as the apomorphine-induced verticalization test or the 5-HTP-induced hyperkinesis test. As a result, a large number of experimental animals and groups is often required. In addition, some drugs may “mask” specific behavioral phenomena, leading to false conclusions about the pharmacological activity of the studied compound. A similar problem is observed when using motor tests to assess the neuroprotective activity. A drug may affect the motor activity of the animal with CNS damage, while it may be considered to be an effective neuroprotective agent according to formal criteria such as the length of the distance traveled in the testing time or the mean velocity of movement of the injured animal. Real-time PCR, Western blotting, enzyme immunoassay, etc. involve the collection of biological material, and, therefore, it is practically impossible to carry them out over time for the same specific animal. In addition, these methods can be quite expensive due to the need to purchase consumables and have certain requirements for the conditions of the preparation and storage of biological materials, as well as the fact that several analyzed genes and proteins do not always allow us to tell exactly the complex nature of the drug action.

In connection with the above, the question arises: can neurophysiological methods (in particular, EEG) be a possible link between traditionally used behavioral tests and modern molecular genetic methods? Especially, how relevant can this approach be when studying the pharmacological activity of new drugs?

More than 100 years ago, almost immediately after the possibility of recording the bioelectrical activity of the brain of humans and animals, studies began on the effect of drugs on the parameters of the recorded signal [1]. This was the beginning of pharmacoencephalography, or pharmaco-EEG—a field of research combining electrophysiology and pharmacology, which studies the effect of drugs on the central nervous system by analyzing qualitative and quantitative changes in EEG parameters in response to drug administration. Over time, the method began to find practical application, for example, in the study of new antiepileptic drugs [2], the effects of sleeping pills [3,4], and many other areas of neuropharmacology [5,6,7,8,9].

The use of pharmaco-EEG in clinical practice is still in its infancy, as its disadvantages are considered to be the low translational potential of preclinical studies, the lack of standardization of EEG recording conditions and analysis methods (as a result of which it is difficult to compare the data obtained), interindividual differences, and a large volume of recorded data (which, however, has now ceased to be a problem) [10]. However, recent works have revived interest in this method [11,12], considering pharmaco-EEG to be a promising way to study the brain and create a relevant predictive marker of its reactions to pharmacological agents, since the method is non-invasive and allows the direct observation of the bioelectrical activity of the brain in vivo [13].

Over the past 10 years, two prominent reviews of pharmaco-EEG studies in rodents have been published [14,15]. The first article [15] focused on using the method to evaluate the effects of drugs on sleep parameters and circadian rhythms. The authors also considered the possibility of predicting the activity of pharmacological agents and identifying potential targets based on their effects on sleep. The second review [14] presented models to evaluate the cholinergic, glutamatergic, and dopaminergic effects of drugs. In addition, examples were given of the use of event-related evoked potentials (P50, P300, and mismatch negativity) in preclinical pharmaco-EEG studies.

This paper provides an overview of the main and most promising areas of applications of the method for assessing the specific activity of neurotropic and psychotropic molecules. In contrast to the reviews by Drinkenburg et al. [14,15], the attention is focused on EEG studies of antiepileptic and neuroprotective drugs, as well as the possibility of classifying psychoactive drugs based on their effect on the amplitude-spectral characteristics of the signal. Of course, the scope of application of the method is not limited to these three tasks; therefore, other areas are presented, that, in our opinion, may also be of interest to researchers. In the conclusion, we present considerations about the possible prospects of each direction, the advantages and disadvantages of pharmaco-EEG, and some recommendations for working with the method. The main attention was focused on works in which the authors used rats as a model subject. However, most of the research problems presented in this review are solved both when working with mice and other small laboratory animals. In general, the present review is addressed mainly to pharmacologists or neuroscientists involved in the study of the pharmacological activity of new molecules, and, therefore, the main emphasis is placed on the possible applications of ECoG as a tool to complement behavioral and molecular research methods.

## 2. Electrodes for Recording of Brain Bioelectrical Activity

In experiments on rodent, electrodes are placed on the surface of the skull (electroencephalography), in the cerebral cortex epidurally and subdurally (electrocorticography), or introduced into brain structures to record local field potentials (electrosubcorticography). These recordings reflect the total electrical activity originating from extracellular excitatory and inhibitory currents, for example, in the dendrites of cortical pyramidal cells [16]. Another source may be underlying structures, as evidenced by the activation of theta activity in the cerebral cortex during exploratory behavior in rats, even though these waves are mainly of hippocampal origin [17,18]. In experimental practice, the corticographic recording of bioelectrical potentials of the brain is the most common; therefore, in this review, the majority of the presented works are ECoG studies.

Many manufacturers of neurophysiological equipment offer a wide selection of electrodes, from standard needle or concentric to flexible microelectrode arrays. Due to the fact that most of them are produced abroad and have high costs, researchers often make the configurations of interest themselves, using simple and affordable materials [8,19,20] (Figure 1A). Electrodes are usually based on biologically inert, non-oxidizing metals such as stainless steel [21], nichrome [20], or platinum [22]. The insulation of non-recording areas is carried out using various varnishes [23], films [24], or heat shrink tubes [20], but, in some cases, it may be absent, for example, when using screw [25] or needle electrodes [26] (Table 1).

Typically, in rodents, the area of implantation is limited to the dorsal surface due to the large lateral temporal muscles on both sides. However, to record acoustic evoked potentials, it is possible to immerse electrodes in the auditory cortex after first removing the temporalis muscle [27]. The registration of bioelectrical activity of the brain from the surface of the skull is less invasive and optimal for newborn rats with thin cranial bones. However, the recorded signal compared to, for example, ECoG, is “noisier” and more vulnerable to motion artifacts. Generally, superficial designs on the skull surface are short-lived and not suitable for chronic experiments. Therefore, when the periodic testing of animals is necessary, epidural and subdural electrodes are used, the most popular option being stainless steel screws. They can be used to record the activity of several parts of the cortex in both hemispheres at once, give a good signal-to-noise ratio, and last for many months. Moreover, these electrodes are usually not insulated and their implantation does not require any advanced surgical skills.

Needle and wire electrodes have some advantages over screw electrodes since they have a smaller diameter, and, due to this, a more localized signal recording is possible. They are also usually less likely to compress and damage brain tissue; however, the puncturing of the dura mater causes areas of hemorrhage, and, if aseptic measures are insufficiently observed, infectious inflammation is possible [28]. Due to the relative simplicity of manufacturing and the possibility of placing 6–10 or more electrodes on the skull, this type of electrodes is the most optimal if it is necessary to register the localized activity of several cortical areas bilaterally.

The most high-tech type of ECoG electrodes are electrode matrices consisting of a soft polymer base (e.g., silicone) and electrode channels incorporated into it. Such matrices are produced on a 3D bioprinter by robotically depositing low-viscosity electrically conductive inks and extruding insulating silicone pastes [22]. Ideally, such implants should be flexible and stretchable, and match the mechanical properties of natural meninges [29]. As a rule, electrode arrays are placed on the surface of the dura mater under the skull bones after preliminary trepanation. The high density of recording areas makes it possible to study in detail the work of different parts of the cerebral cortex. Tasnim et al. proposed a simple and low-cost technique of ECoG array fabrication from thin wires and Parafilm without the need for 3D printing [24]. Despite the obvious advantages in signal registration, microelectrode arrays as a technical solution also have some shortcomings. First of all, in most cases, such electrodes are difficult to fabricate, and require specialized equipment (e.g., 3D bioprinter), trained personnel (usually a collaboration of chemists and bioengineers), and expensive consumables, which makes them inaccessible to most laboratories. In addition, matrix implantation usually requires two trepanation windows through which the matrix is placed on the cortical surface and subsequently fixed [22]. This increases the requirements for the level of surgical personnel and the duration of postoperative follow-up. In general, such electrodes are usually used for studying the delicate mechanisms of cortical neuron functioning, for example, in the realization of sensorimotor functions [22,24,29].

Implantable electrodes are typically combined into one common connector after manufacture to facilitate the connection of the test animal to the recording device (Figure 1B). Such connectors can be BLD, BSD, or other sockets. Regardless of the chosen type of ECoG electrodes, when recording a signal in awake, freely moving animals, a design is required that protects the recording electrodes and attaches them to the skull of rodents. This is usually achieved by screwing fixing screws (screw electrodes perform this function independently) along the edges of the dorsal surface of the skull and covering the entire structure with dental resin.

**Table 1 brainsci-14-00772-t001:** Examples of electrodes for recording ECoG in rats in neurobiological studies.

Electrode Type	Materials Used	Insulation Type	Experimental Tasks
Screw electrodes	Stainless steel	None	Studying the rhythms wakeful rest, slow-wave, and REM sleep at different brain areas in rats [21]. Detection of interhemispheric symmetry and asymmetry of absence-type spike-wave discharges caused by systemic administration of pentylenetetrazole [25].
Needle electrodes	Stainless steel, silver	None	Study of the motion of epileptiform activity wave on a model of cortical epilepsy [26]. Assessment of age-related dynamics of ECoG and seizure activity of Wistar rats in a cortical model of focal epilepsy [19,30].
Wire electrodes	Nichrome	Shrink tube	Analysis of changes in amplitude spectral characteristics, coherence, VEP, and SSEP in rats after traumatic brain injury [31,32,33,34].
	Platinum/iridium (80%/20%)		Evaluation of the effect of clozapine and the 5-HT2A-antagonist RU-31 on the power spectral density of EEG rhythms in rats using a model of schizophrenia with neonatal destruction of the ventral hippocampus [8].
Electrode arrays	Aluminum (2.5%), chromium (20%), copper (2.5%), and nickel (75%)	Parafilm	Registration of activity of motor and somatosensory cortex during locomotion on a treadmill [24].
	Platinum	Silicone	Registration and analysis of spectral power during standing and walking, MEP and SSEP [22].

Note: REM—rapid eye movement, VEP—visual evoked potential, MEP—motor evoked potential, SSEP—somatosensory evoked potential.

## 3. Basic Methods of Analyzing Bioelectrical Activity of the Brain

### 3.1. Spectral Analysis

One of the main advantages of EEG over other neuroimaging methods based on the assessment of hemodynamic or neurochemical processes, such as fMRI and PET, is its high temporal resolution, which allows for the study of neuronal activity in ranges of tens of milliseconds. Thus, a large amount of useful information about the functioning of the brain can be obtained using Fourier transform and signal decomposition into frequencies (so-called spectral analysis).

A spectral analysis of EEG allows the quantitative assessment of rhythmic activity of a specific frequency. Based on numerous studies that have reported significant associations between the EEG spectrum and behavior, cognitive status, or mental illness, EEG spectral analysis is now accepted as one of the main analysis methods in the field of neurobiology. Currently, in clinical practice and in animal studies, δ- (“delta”, 1–4 Hz), θ- (“theta”, 4–8 Hz), α- (“alpha”, 8–12 Hz), β- (“beta”, 12–30 Hz), and γ- (“gamma”, 30–100 Hz) rhythms are distinguished. These frequency ranges may vary from study to study and from laboratory to laboratory, but it is generally accepted that small differences (less than 1 Hz) are not significant [35]. It is generally accepted that the low-frequency range reflects subconscious states and is predominant during deep sleep or coma, whereas higher frequencies are associated with active states or cognitive functions [36].

Graphically, the results of the spectral analysis can be presented in the form of power spectral density plots (spectrograms) for each lead or in the form of topographic maps. When plotting spectrograms, the abscissa axis represents frequencies, and the ordinate axis represents power spectral density values characterizing the expression of each frequency component in the analyzed EEG fragment. For each frequency range (δ-, θ-, α-, β-, or γ-), the absolute (AP) or relative power (RP) can be calculated. In the former case, the AP value (μV^2^) is calculated as the area under the corresponding area of the spectrogram over the selected rhythm. The RP value, or power index (%), is the ratio of the area under the corresponding section of the spectrogram to the total area of the selected frequency ranges, multiplied by 100%.

It is well-known that changes in the EEG power spectrum are directly or indirectly related to a variety of ongoing brain activities. For example, in rats, the activity of the θ rhythm (which is hippocampal in nature [17,18]) increases during exploratory activity or during the REM phase of sleep [37], and the predominance of δ waves (dominant in the frontal leads) can be observed during NREM sleep [38] (Figure 1C). More detailed information on behavioral correlates and the nature of the recorded rhythms in mammals can be found in the review by Hernan et al. [39].

### 3.2. Connectivity Analysis

Since behavioral responses and, in particular, cognitive functions require the coordinated work of various parts of the brain, its quantitative analysis is of great research interest. Functional connections between pairs of brain leads can be identified using a cross-correlation and coherence analysis; however, there are other methodological approaches such as the phase locking value (PLV) and mutual information (MI) functions, phase lag index (PLI), and Granger causality test [40,41]. The basic method is a correlation analysis, which has been used since the first paper EEGs were analyzed when calculations were performed manually. The cross-correlation function reflects the degree of coupling of processes at different points in the brain by identifying periodic components common to two EEGs. The degree of correlation is expressed by a correlation coefficient ranging from -1 (EEGs are out of phase) to +1 (EEGs are identical) [42]. With the development of computer technology, correlation analysis has been significantly supplanted by coherence analysis, which makes it possible to determine the degree of similarity of oscillatory electrophysiological processes of two or more brain regions within specific frequency ranges [43]. A functional connectivity analysis is used in the diagnosis of various neurological and mental diseases. Recent studies have shown impaired or abnormal connectivity patterns in patients with schizophrenia [44,45], cognitive impairment in Parkinson’s disease [46], and post-traumatic stress disorder [47,48], and after traumatic brain injury [49]. In particular, the functional connectivity analysis has proven useful for studying epilepsy, a disease accompanied by severe disturbances in the functioning of neuronal networks of the brain [50].

### 3.3. Evoked Potentials

The study of brain evoked potentials (EPs) is based on recording EEG/ECoG signal responses to external stimuli such as somatosensory stimulation, photostimulation, or auditory signal or internal events associated with cognitive activity requiring recognition, decision making, or the initiation of a motor response [51]. Visual evoked potentials (VEPs) are responses of the visual cortex to visual stimulation. They can be initiated by flashes of light or a specific pattern such as a checkerboard and are recorded from the occipital electrodes. Somatosensory evoked potentials (SSEPs) are EPs recorded in the brain or spinal cord during the current stimulation of peripheral nerves or muscles. When sound stimuli are applied to the temporal cortex, auditory evoked potentials (AEPs) are generated. Regardless of the nature of the stimulus, the reflex response of the cortex is a curve consisting of several positive and negative peaks (Figure 1D). The analyzed quantitative indicators of EPs are the latencies and amplitudes of peaks, as well as the duration and amplitude of interpeak intervals. It is generally accepted that the increase in the latency of recorded responses is associated with demyelination processes, for example, in multiple sclerosis [52]. In organic brain lesions, for example, due to traumatic brain injury, tumor, or stroke, the amplitudes of EP peaks may decrease until they are completely absent [53,54,55].

## 4. Application of ECoG in Neuropharmacological Research

### 4.1. Study of Antiepileptic Activity

Epilepsy is one of the most common neurological disorders, affecting more than 50 million people worldwide [56]. Since EEG allows a direct “look” at the bioelectrical activity of the brain, this method is key to diagnosing epilepsy in clinical practice. However, the quantitative assessment of EEG changes on the background of pharmacotherapy has not yet become widely used in neurological practice and remains, for the most part, within the framework of initiative projects. A review by Höller et al. [57] analyzed 37 studies in which the authors used Pharmaco-EEG to evaluate the antiepileptic activity of drugs. In most cases, the effects of drugs on the spectral characteristics of the signal were studied, their effect on cognitive functions was identified using event-related evoked potentials, and attempts were made to predict the effectiveness of treatment. Despite the fact that the groups included a relatively small number of patients (usually no more than 20), the authors of the review talk about the importance of using the method in studies with patients with epilepsy, citing as arguments the ease of its inclusion in the patient management protocol and the emergence of unique algorithms for analyzing the bioelectrical activity of the brain, as well as new antiepileptic drugs. Some optimism is inspired by the recent work of Ricci et al. [12], which assessed the value of a spectral analysis and calculation of EEG connectivity for predicting the outcome of the treatment of temporal lobe epilepsy with Levitiracetam. An analysis of the quality of the ROC-curve classification showed a prediction accuracy of about 60–90% based on the values of the PLV connectivity of the δ-, θ-, α-, β-, and γ-rhythms.

Despite the availability of a large number of antiepileptic drugs with different mechanisms of action [2], to date, about 20–30% of patients with epilepsy do not respond to pharmacotherapy [58]. Taking into account some obvious differences in the brain neuroanatomy of mice, rats, and humans, their EEGs are, in many respects, still similar in their amplitude-spectral characteristics and manifestations of pathological processes, including epiactivity [59]. Commonly known risk factors for epilepsy in humans are cerebrovascular diseases, brain tumors, alcohol, traumatic brain injury, disorders of cortical formation in the embryonic period, genetic disorders, neuroinfections, and many others. Therefore, it is particularly important to note that, to date, many models of this disease in rodents have been proposed which focus on one or another of the above factors [2]. As in many other areas of neuroscience, ECoGs allow us not only to study the etiology and pathogenesis of epilepsy, but also to objectively assess the effectiveness of antiepileptic drugs [60,61,62]. In connection with the above, it is not surprising that, when searching for works using ECoG in rodents, one can notice that a significant part of them is devoted to the study of the antiepileptic activity of various pharmacological agents (Figure 2).

According to the Antiepileptic Drug Development (ADD) program, established in 1975 in the USA, the study of the antiepileptic activity of a putative candidate molecule should begin with screening studies in three experimental models in mice and rats. These include models of primary generalized epilepsy in the maximum electric shock (MES) and antagonism tests with corazol, as well as an assessment of neurotoxicity (by deterioration of motor function) in the rotarod test. Over time, the 6 Hz test and a test using the lamotrigine-resistant kindling rat model of partial seizures were added to these basic methods. Since 2015, when the Anticonvulsant Screening Project (ASP) was renamed as the Epilepsy Therapy Screening Program (ETSP), one of the main goals of the program has been the search for compounds effective against drug-resistant forms of epilepsy. The research began to be divided into the phases of “identification” and “differentiation”. The first phase begins evaluating the compound in two acute mouse models: The MES test and the 6 Hz partial seizure test, which is conducted at a current of 44 mA (at which many drugs already in clinical use do not suppress seizures). The differentiation phase consists of three tests: an intrahippocampal kainate model of mesial temporal lobe epilepsy in mice, a lamotrigine-resistant amygdala kindling model in rats, and a rat model of chronic epilepsy developing after the induction of status epilepticus by kainite [2].

The advantage of the first model is the occurrence of frequent spontaneous electrographic seizures in the area of kainate administration, which allows for the testing of drugs in a short period of EEG recording [63]. When status epilepticus is induced by the systemic administration of kainate, more widespread and bilateral neuronal damage occurs, especially in limbic regions. Unlike the intrahippocampal model in mice, in this case, spontaneous recurrent seizures are unstable, which is why long-term 24 h video-EEG monitoring is required to correctly assess antiepileptic activity [63].

The study of epilepsy in animal models has advanced significantly since researchers began simultaneously recording EEG and video signals in test animals. With this approach, an accurate correlation between a single event and a phenomenon on the encephalogram became possible. Viewing simultaneous EEG and video data may also be useful in confirming that the observed behavior is an epileptic seizure and not a specific behavior (e.g., scratching). In addition, video-EEG monitoring helps to classify seizures (e.g., with or without seizures, absence seizures, and partial or generalized forms) and determine their temporal course. During interictal periods, video recording is necessary to assess the animal’s activity (e.g., resting, exploring, locomotion, or grooming) to better correlate with the EEG. With the use of a special video camera or infrared illumination, video monitoring is possible even in the dark [64].

In the Russian Guidelines for Preclinical Studies [65], EEG recording is performed using a model of primary generalized epileptiform activity induced by bemegride or corazol. Experiments are carried out on rats with electrodes chronically implanted in the sensorimotor cortex and some subcortical structures, for example, in the dorsal hippocampus or the amygdala complex. The recording of electrical activity usually occurs while the animal is freely moving around the chamber. The test substance is administered several minutes before the administration of bemegride (or corazol) so that their peak activity coincides with the peak activity of the compound that causes seizures. The anticonvulsant effect of the substances is assessed by changes in the number of discharges per minute, the duration of individual discharges, and the total duration of discharges per minute. It is assumed that different mechanisms underlie the effect of substances on the number of discharges and their duration, and antiepileptic effects can be expressed both in a decrease in the number of discharges, and in a decrease in their duration, or in a combination of these effects.

Another model for which EEG registration in rats is used is the technique of an epileptogenic chronic focus caused by the application of cobalt to the surface of the sensorimotor cortex [65]. Simultaneously, the animals are implanted with electrodes in the ipsilateral and contralateral (relative to the focus) zone of the sensorimotor cortex and some structures of the limbic–hypothalamic complex. The bioelectrical activity of the studied structures is recorded in freely moving rats daily starting from the second day after application. A daily EEG recording allows us to identify the features of the development of the epileptic system at various times after the application of an epileptogen and to select the most informative periods for exposure to antiepileptic drugs. The number and duration of discharges per minute and the duration of one discharge, as well as the latent time of occurrence of individual paroxysms in each structure under study are counted, which makes it possible to determine the most accurate location of application of the substance and make an assumption about the possible mechanism of action of the substance.

The ECoG analysis in rodent models includes an assessment of signal changes during a seizure (ictal activity), interictal sharp waves or spikes, and non-epileptiform (background) activity. Spikes and sharp waves can be identified visually, but counting them manually can be time-consuming, so approaches based on machine-learning algorithms have now been proposed [66,67,68]. Since “epilepsy is more than just seizures”, a detailed analysis of the background EEG activity in animal models provides significantly more information than is provided by the simple counting of spikes and sharp waves. The elementary spectral analysis can be an important starting point, but, when limited to predefined frequency intervals within a given frequency range (e.g., 1 to 35 Hz divided arbitrarily into delta, theta, alpha, and beta bands), some important data may be lost. Filtering > 80 Hz allows high-frequency oscillations (HFOs) of 80–500 Hz to be analyzed. These waves are recorded in the temporal lobe regions of animals with epilepsy during interictal periods, but they also occur before seizure onset and during the ictal period. Moreover, HFOs change during different stages of epileptogenesis, which some authors attribute to neuronal death and the subsequent neuroplasticity [69,70]. It can be assumed that studying the mechanisms of high-frequency waves is the key to unraveling the pathogenesis of seizures. This is fueled by the fact that, although most antiepileptic drugs are able to reduce the number of spikes and sharp waves, virtually none of them have an effect on the HFO. Thus, it can be inferred that available antiepileptic drugs reduce only ictogenesis in animal models, but they cannot affect the cellular mechanisms underlying the pathogenesis of the disease.

As part of EEG studies of new antiepileptic drugs in rodents, laboratories use their own individual methodological approaches. Due to the lack of uniform requirements for conducting such experiments, the American Epilepsy Society (AES) and the Translational Working Group of the International League Against Epilepsy (ILAE) have published guidelines [28,39,71,72] for EEG studies in rodents. These publications raised important issues, the solution of which, on the one hand, will help improve the quality of the experimental data, and, on the other hand, with the standardization of protocols, will make it possible to compare the results obtained in different research groups. The list of issues raised includes the size, materials, and implantation depth of the recording electrodes, the choice between wired and wireless recording methods, recommended filters and sampling rates, and the nature of emerging artifacts and methods for their detection, as well as many others (Table 2).

### 4.2. The Use of ECoG in the Valuation of Neuroprotecive Activity

A major challenge in developing new drugs aimed at increasing neuronal survival in neuroinjury, ischemia, or neurodegeneration is the low translational potential of experimental animal studies [73,74,75]. The development and validation of new approaches to assessing the neuroprotective activity of potential drugs may be one of the possible solutions to this problem. To date, in experimental practice, the use of behavioral and functional tests, methods of biochemical and immunohistochemical studies, is an unspoken standard for evaluating the effectiveness of pharmacological agents designed to correct neurological disorders in CNS damage. Neuroimaging methods such as magnetic resonance, computed tomography, and positron emission tomography are used much less frequently due to their high cost, as well as the small size of laboratory animals, which places high demands on the resolution capability of the devices used.

Few studies have been published in which EEG/ECoG has been used to assess the neuroprotective activity of drugs. Meanwhile, it is worth noting the undoubted advantages of using these methods in such studies. First of all, it is possible to assess not only the functional state of individual areas of the brain (for example, when conducting an amplitude-spectral analysis), but also the state of the inter- and intrahemispheric connections (in the case of the cross-correlation and coherence analysis). An analysis of evoked potentials makes it possible to evaluate the functioning of sensory systems during photo-, audio-, or somatosensory stimulation. It is also important that the registration of brain bioelectrical activity can be performed repeatedly, thereby observing the state of the test animals in dynamics.

ECoG is a fairly sensitive method for detecting functional disorders after a traumatic brain injury [31,32,33,34,76] or ischemic stroke [77,78,79,80], as well as in neurometabolic disorders [81,82,83,84] or neurodegenerative processes [85]. Table 3 summarizes the main changes in the characteristics of bioelectrical activity of the brain observed in rats when modeling such pathologies.

In general, organic brain lesions in rats, regardless of etiology, cause rather general changes in the amplitude and spectral characteristics of ECoG. As a rule, these are increased power of delta rhythms, and decreased interhemispheric and intrahemispheric connectivity, as well as decreased amplitudes and increased latency of SSEPs and VEPs. The review by Moyanova and Dijkhuizen considered in more detail the EEG phenomena detected at different periods in rats after an MCA occlusion [86]. Surprisingly, certain parameters of bioelectrical activity of the brain in rats can correlate with functional recovery in rats after a spinal cord injury. In the work of Pu et al. [87], such parameters were the sample entropy, detrended fluctuation analysis (DFA), and Kolmogorov complexity—nonlinear dynamic metrics reflecting the complexity of the EEG signal. On the first day after the injury, there was a pronounced decrease in it, but, over the following days, the values of all three parameters returned to the initial values before the injury. It is important that the observed dynamics correlated with the BBB scale scores, which allowed the authors to assume that the metrics used reflect the processes of neuroplasticity.

In the study by Gantsgorn et al. [78] on a model of cerebral ischemia in rats, a positive effect of the preventive administration of combinations of melatonin with vinpocetine or piracetam (M + V and M + P, respectively) was shown on the parameters of the EEG rhythms of animals that survived the bilateral occlusion of the common carotid arteries. The specific effect of the studied combinations used for 14 days on the relative spectral power (RSP) values of separate frequency ranges was revealed, expressed to varying degrees in the recorded cortical (somatosensory cortex) and subcortical leads (CA1 zone of the hippocampus). The M + P combination had a greater effect on the fast-wave bioelectrical activity of the brain in rats, stimulating it before and after ischemia. The M + V combination, in turn, significantly reduced the slow-wave activity of the δ rhythm in all leads but increased the RSP of the θ rhythm in the hippocampus. Only this group showed a frequency range close to the background. Although the power indices of the θ-, α-, and β-rhythms were slightly lower than their initial values, the θ-frequency range dominated over the δ-activity, indicating the awake state of the brain of rats in this group. The RSPs of the θ- and α-rhythms had similar values, with the δ-activity being lower and the β-rhythm being slightly higher compared to the group of sham-operated rats.

In our experiments, we evaluated the effect of the alpha-2 adrenoreceptor agonists mafedine and dexmedetomidine on the parameters of bioelectrical activity of the brain of rats after a traumatic brain injury (TBI) modeled by the controlled cortical impact injury (CCI) method [34]. This group of drugs is one of the most promising among the means of correction of a neurological deficit, which is supported by the data of meta-analysis, including nine randomized placebo-controlled trials involving 879 patients who suffered an ischemic stroke [88]. The alpha-2 adrenergic agonist dexmedetomidine, used in anesthesiology, has shown the ability to reduce the release of pro-inflammatory mediators and neuroendocrine hormones, maintain intracranial hemostasis, and reduce the amount of brain damage during ischemia. The neuroprotective activity of another agent from this group, mafedine, was shown in a model of TBI in rats [89]. Its 7-day course of administration led to a decrease in the volume of brain damage and a decrease in the intensity of inflammatory processes in the area of injury while improving the general neurological status in injured animals.

To assess the recovery of rats after TBI, ECoG was recorded from electrodes implanted bilaterally in the area of the secondary and primary motor cortex and primary somatosensory cortex above the hippocampus. Additionally, the visual (VEP) and somatosensory (SSEP) evoked potentials were analyzed. It was found that the administration of mafedine 1 h after TBI and in the next 6 days led to the normalization of the interhemispheric connections of brain regions remote from the area of injury, as well as the intrahemispheric connections of the healthy hemisphere by day 7 after injury. In addition, positive changes in cortical responses to photo- and somatosensory stimulation were noted in such animals. In rats injected with mafedine, on day 3 after surgery, the amplitude of the P2 peak of the VEP in the C4 and O1 leads was significantly lower (*p* < 0.01 and *p* < 0.05, respectively) than the values of untreated animals, approaching the control values. Despite the fact that both drugs did not change the parameters of the SSEP curves of lead C3 evoked during the stimulation of the right sciatic nerve (contralateral responses), the mafedine group showed an increase in the latency of the early and late responses (P2, N2, P3, and N3) in lead C4 (ipsilateral responses) compared to control animals (*p* < 0.01, *p* < 0.05) [34].

### 4.3. Pharmacoencephalography as a Classification and Prediction Tool

Attention to Pharmaco-EEG as a tool for studying psychoactive drugs has waxed and waned several times over the past 70 years [15]. The first “outbreak” occurred in the 1950s to 1960s of the last century, when antipsychotics appeared—phenothiazine derivatives and tricyclic antidepressants. Their EEG analysis became one of the methods for assessing the clinical effects, and, in addition, attempts at translational research from animals to humans began immediately. Most experiments at that time were carried out on large laboratory animals such as rabbits [90], cats [91], dogs [92], and monkeys [93], while work on rodents was rare. Due to the limited computational resources of that time, the work was focused more on the search for amplitude-spectral phenomena than on classification and prediction problems. The first attempts to classify the effects of new drugs were based on a visual search for the similarity of their EEG effects with those of known drugs. Of course, this approach directly depended on the experience and attentiveness of a researcher, and, therefore, was not considered objective in the scientific community.

In 1972, Itil et al. used a training sample of anxiolytics, psychostimulants, neuroleptics, and antidepressants to determine by a discriminant analysis that GB-94, developed initially as an antihistamine anti-inflammatory drug, might have an antidepressant effect [94]. Subsequently, in clinical trials, GB-94, later named mianserin, confirmed this prediction, thus sparking interest in pharmaco-EEG as a tool for predicting pharmacologic activity. At that time, there were no relevant models of mental illness, so the proposed approach, adapted for laboratory animals, became one of the first tools of translational research, long before the latter became key in drug development. At the same time, many researchers began to switch to rats because of their lower cost and ease of maintenance and handling. By the late 1980s and early 1990s, with the assistance of pharmaceutical companies, research groups began to appear creating libraries of EEG effects of psychoactive drugs to predict the pharmacological activity of new compounds [95,96,97,98].

The most successful and long-lived was the research group of Wilfried Dimpfel, whose first results were published in the mid-1980s and continue to be published to this day. A 1985 paper [97] used discriminant analysis to distinguish the effects of sulpiride, clozapine, and haloperidol by comparison with the effects of amphetamine, diazepam, imipramine, and chlorpromazine in rats. The ratios before and after drug administration of the absolute powers of the δ-, θ-, α-, and β-rhythm of signals recorded bilaterally in the areas of projection of the sensorimotor cortex, striatum, and reticular formation were used as the input data. Subsequent work was devoted to recording the effects of various groups of drugs, which later allowed the authors to compile a library of “electropharmacograms” [99], including antipsychotics, opioid and non-opioid analgesics, antidepressants, psychostimulants, sedatives and anticonvulsants, and even hallucinogens (LSD, MK 801, etc.). From 2009 to the present, the library has been actively used to study the effects of plant extracts. For example, the antidepressant effect of the flavonoids rutin and quercetin [100] and a number of other compounds was predicted, and a variant of the pharmacological classification of plant extracts based on their electropharmacograms was proposed [5].

A similar approach was developed in 1993 by Dutch researchers Krijzer et al. [101]. The bioelectrical activity of the brain was recorded from the frontal and parietal areas of the cortex. The power ratio values of 256 spectral ranges from 0.36 to 100 Hz before drug administration and 20 and 45 min after were used as the input data for further analysis. The analysis of variance, *t*-test, and subsequent normalization by degrees of freedom transformed the obtained values of the primary analysis into so-called n-profiles. A further discriminant analysis made it possible to compare the n-profiles of drugs from different groups and to distinguish between the effects of antidepressants, antipsychotics, anxiolytics, and psychostimulants. The proposed method made it possible to predict the antidepressant and anxiolytic effects of E-10-hydroxynortriptyline, the active metabolite of nortriptyline, which confirmed previous clinical observations in patients with depression [102].

However, despite the interesting results obtained, the approach was not developed and the above study is the last published work that used n-profiles to predict pharmacological activity. In addition, due to individual failures in predicting the pharmacological activity using the pharmaco-EEG method [103], the emergence of valid models of mental illness in rodents, and molecular research methods (RT-PCR, and Western blotting), as well as neuroimaging techniques (e.g., functional MRI and PET), interest in the method decreased in the 1990s.

Our research group conducted a series of experiments using the naïve Bayes classifier (NBC) as a tool for classifying and predicting the pharmacological activity of psychoactive drugs based on their effect on electrocorticogram parameters in rats [9,104,105]. The NBC is a simple probabilistic classifier in which each parameter of the data being classified is considered independently of other characteristics. It is widely used in medical practice, for example, for predicting drug resistance in chemotherapy [106,107], diagnosing diseases [108,109], or assessing the risk of drug toxicity [110,111]. Of course, the possibilities of using this algorithm are not limited to the above examples, and, every year, more and more papers appear where the authors successfully use this approach in classification and prediction problems.

All experiments (Figure 3) were performed on white outbred male rats with chronically implanted ECoG electrodes bilaterally in the primary motor cortex, sensory cortex over the hippocampus, and secondary motor cortex [20]. The signal was recorded simultaneously with the video recording of behavior in a home cage under artificial lighting. The recording duration was 1 h and included 30 min of background activity (before the injection of the drug or saline) and 30 min after the injection. Two 60 s sections of the recording were taken for further analysis: immediately before administration and 20 min after. The selected fragments contained ECoG recordings in a quiet awake state, since locomotion, stance, grooming, or scratching can significantly hinder the detection and discrimination of drug effects on the brain bioelectrical activity in rats [112].

The ECoG analysis included the calculation of 132 indicators of amplitude-spectral characteristics of the signal (average amplitudes, indices and powers of δ-, θ-, α-, and β-rhythms, etc.). For each recording, parameter values were calculated before and after drug administration. Data dimensionality reduction was carried out using the principal component analysis (PCA), a multivariate statistical analysis technique used to reduce the dimensionality of the feature space with a minimal loss of useful information. The essence of the method is an orthogonal linear transformation that maps data from the original feature space into a new space of lower dimensionality. In this case, the first axis of the new coordinate system is constructed in such a way that the dispersion of the data along it is the maximum. The second axis is constructed orthogonally to the first one so that the dispersion is the maximum of the remaining possible ones, etc. The first axis is called the first principal component, the second axis is called the second principal component, etc. [113]. Next, using the calculated values of the principal components for each record, the pharmacological activity was classified based on the probability of similarity to a particular drug from the training set.

In the first experimental series [105], a library of ECoG recordings was generated during the administration of several typical and atypical antipsychotic drugs, chlorpromazine, haloperidol, droperidol, tiapride, and sulpiride, which were used as a training set. Additionally, the ECoG effects of the tricyclic antidepressant amitriptyline, the acetylcholinesterase inhibitor galantamine, and the benzodiazepine anxiolytic phenazepam as reference drugs were recorded. The use of the PCA and NBC made it possible to identify the specific effect of antipsychotic drugs on the parameters of bioelectrical activity of the brain in rats, differentiating them from saline solution (control), as well as the anxiolytic with the sedative effect—phenazepam. The similarity of the effects of antipsychotics with similar chemical structures was noted: butyrophenone derivatives—haloperidol and droperidol, as well as substituted benzamides—tiapride and sulpiride (Figure 3B).

Later, the sensitivity of the proposed method was assessed to identify the dose-dependence of the effects of antipsychotic drugs [104]. Two drugs with antipsychotic action in three doses were chosen as the drugs studied: chlorpromazine (0.1, 1.0, and 10 mg/kg) and promethazine (0.5, 5.0, and 20 mg/kg). The training set, against which the pharmacological effects of the studied drugs were determined during the work, were the dopamine D2-receptor blocker haloperidol, the M-cholinoblocker tropicamide, and the H_1_ receptor antagonist chloropyramine, as well as the α_2_-adrenergic agonist dexmedetomidine and the GABA mimetic phenazepam. This work showed that the lowest dose of chlorpromazine (0.1 mg/kg) can be characterized as antipsychotic with a pronounced histaminolytic effect, while the highest dose (10 mg/kg) demonstrates a predominantly antipsychotic effect with a cataleptogenic effect. All three doses of chlorpromazine exhibited anticholinergic effects, which increased with increasing dose. Promethazine showed a clear dose-dependent transition from antipsychotic to cataleptogenic action, as well as a pronounced M-anticholinergic effect in all administered doses (Figure 3C).

Eventually, we assessed the possibilities of a combination of the PCA and NBC as a screening tool for new little-studied molecules [9]. The object of the study was the amino ester of valproic acid (AVA), which exhibits the properties of an antidote for acute poisoning with anticholinesterase drugs [114]. AVA was administered at doses of 0.5, 5, and 30 mg/kg. The effects matrices of seven drugs were used as a training set, relative to which the effects of each dose of the compound under study were classified: the antiepileptic drug sodium valproate, the dopamine D2 receptor blocker haloperidol, the M-cholinoblocker tropicamide, the H_1_ receptor antagonist chloropyramine, the acetylcholinesterase inhibitor galantamine, the sedative dexmedetomidine, and the anxiolytic phenazepam. It was found that AVA at a dose of 0.5 mg/kg exhibits effects similar to those of sodium valproate, and a tenfold increase in the dose leads to a predominance of an atropine-like effect (Figure 3D). When administered at a dose of 30 mg/kg, the compound exhibits dexmedetomidine-like effects. The ability of AVA to block central M-cholinergic receptors was confirmed by the arecoline test, in which the substance at a dose of 88 mg/kg completely abolished the onset of tremors in mice. The dexmedetomidine-like action was blocked by the administration of atipamezole in equimolar amounts, which may suggest the ability of AVA at high doses to activate central α_2_-adrenoreceptors. The results of molecular docking confirmed the likelihood of AVA binding to all three subtypes of α_2_-adrenoceptors; moreover, the binding was specific to the original amino ester molecule and not to its probable active metabolites. This, to some extent, explains why the “dexmedetomidine-like” effect of AVA was found in high doses [9].

Based on the results obtained, we can conclude that the combination of the PCA and NBC is relevant for the assessing of the pharmacological activity of compounds based on their effect on the amplitude-spectral characteristics of ECoG in rats. At present, it seems to us that a revision of the method as a tool of pharmacological screening is necessary. This is primarily facilitated by both the emergence of new selective agonists and antagonists of receptors of mediator systems, and the discovery of new systems themselves [115,116]).

The failures of past years could largely be due to the fact that psychoactive drugs available at that time, which did not have a sufficient selectivity for molecular targets, were used as reference drugs. For example, amitriptyline, used in the 1960s to 1970s as a reference antidepressant, is capable of blocking noradrenaline (NET) and 5-hydroxytryptamine (SERT) transporters, and M-cholinergic and H_1_-histamine receptors [117]. In this regard, a question naturally arose: when similarities in the EEG effects of a new drug with amitriptyline are discovered, what specific pharmacological effect is predicted? However, using only highly selective drugs, apparently, will also not allow us to arrive at an “ideal” training set. For example, the target–pharmacological effect relationship is not clearly demonstrated for all receptors. For many selective agonist–antagonist molecules, the pharmacokinetics has not been studied; in particular, it is not known how well they cross the blood–brain barrier. In addition, over time, the “loss of selectivity” of the reference substance is possible, when new molecular targets are identified for the ligand and it is no longer selective. Therefore, it is likely that the ideal training set, relative to which the pharmacological activity of new unstudied molecules will be predicted, should be a “set” of drugs and selective agonists or antagonists. Another possible option is to use two sets, the first of which uses an “effect-based” classification and the second of which uses a “target-based” classification.

For a greater efficiency of the proposed method, it is necessary to replenish the library of records used with a large number of reference psychotropic drugs, as well as to select the optimal settings of the classification algorithms used. Regardless of the chosen principle of creating a library, future work will need to answer some important questions. First of all, what ECoG characteristics are the most informative in terms of the effects of psychoactive drugs? We use the amplitude-spectral characteristics of the signal, but a considerable amount of information is also contained in data from, for example, the cross-correlation and coherence analysis. However, according to our observations (data not published), cross-correlations and coherence as calculated by Neuron-Spectrum.NETomega do not provide a logical picture when classifying using the PCA and NBC.

Another issue is choosing the optimal number of records in each group. In our work, the NBC is a probabilistic model for which there is no formal requirement for the number of input examples for each class. For its correct operation, the independence of features for each example is important, which, in our case, is guaranteed by the use of principal components, which are independent by definition. Sufficiency strongly depends on the structure of the data and on the variability of features within one class. In order to judge it, in machine learning for solving classification problems, one usually looks at the quality of the classification and the stability of the quality under random repartition (cross-validation). The issue of minimum sufficient samples is complex and, unlike standard statistical tests, does not have an analytical solution that can be obtained using one or another tool for calculating the power of the test. To answer this question when using machine-learning models, we study the behavior of learning curves for different sample sizes, assess the stability of the predictions, and analyze their change with an increasing number of examples in the sample [118].

It is important that recordings of training sets (libraries) of EEG effects of drugs are carried out with repeated use of animals—i.e., the 3R principles (Responsibility, Reusability, and Readability) are followed—which are already a generally accepted basis for biological research. If an effective pharmacologic screening tool is developed, it will be possible to reduce the number of laboratory animals in exploratory studies, especially if libraries and developing algorithms and software for classification and prediction become publicly available. If the interval between testing drugs is greater than four half-lives and if the background activity is recorded before each recording, a sufficiently large number of pharmacologic agents can be tested in a small sample of animals. This approach has been successfully used not only in EEG studies by other authors [99,101], but also outside the framework of Pharmaco-EEG, for example, when assessing antiparkinsonian effects [119] or choosing optimal neuromodulatory drugs after spinal cord injury [120].

### 4.4. Other Applications

#### 4.4.1. Evaluation of Hypnotic Effects

Neurophysiological studies of sleep in rodents represent an important tool for studying the neural mechanisms of this process [121,122], for modeling the pathologies underlying sleep disorders [123], as well as a model for assessing the effectiveness of hypnotic drugs [124,125]. Although human sleep stages are well-defined and there is consensus on the criteria for dividing them, there are no generally accepted guidelines for rodents. Importantly, the general phase structure of sleep, consisting of two alternating states (NREM and REM sleep), is conserved in mice and rats [126,127]. In EEG studies, the level of wakefulness in animals is usually determined using the video recording of behavior [128], the recording of the EMG signal [129], or devices that measure the vertical and horizontal activity [130]. The recording electrodes in mice and rats are usually placed in the frontal cortex to detect slow delta oscillations and sleep spindles (NREM phase), while occipital and hippocampal leads are critical for recognizing theta oscillations (REM phase) [38].

An example of an EEG study of the hypnotic effect of a drug is the work of Winrow et al. [124], which assessed the effect of suvorexant, an antagonist of OX_1_ and OX_2_ orexin receptors. Based on EEG and EMG signals, the authors identified four sleep/wake states (active wakefulness, light NREM sleep, slow-wave NREM sleep, and REM sleep) in rats. Suvorexant was administered for 7 days at doses of 10, 30, and 100 mg/kg. It was found that the administration of the drug led to a dose-dependent increase in the duration of slow-wave sleep (by 10–25%) and REM sleep (by 27–48%). At all doses, suvorexant caused a consistent decrease in the duration of active wakefulness with a corresponding increase in the duration of slow and REM sleep within 2–7 h after administration, reducing the latency to falling asleep by 24–41%. Similar EEG effects in the same study were obtained for dogs and rhesus macaques, confirming the translational potential of the rat model for studying the hypnotic effects of drugs.

#### 4.4.2. Evaluation of Analgesic Activity

A study by LeBlanc et al. [131] demonstrated the effectiveness of quantitative EEG in rats for assessing the effects of analgesics in models of acute, inflammatory, and neuropathic pain. EEG was recorded from the area of the somatosensory cortex (S1) bilaterally, as well as from the prefrontal cortex (PFC). Acute pain was modeled by the intradermal injection of the TRPV1 receptor agonist capsaicin, and inflammatory pain was modeled by the intramuscular injection of Freund’s complete adjuvant into the left hind limb. Neuropathic pain was created by the quadruple left-sided ligation of the sciatic nerve, which, after 2 weeks, led to the gradual development of pain with characteristic behavioral phenotypes. The drugs studied were the cyclooxygenase inhibitor ibuprofen, the voltage-gated calcium channel α2δ-subunit ligand pregabalin, and the voltage-gated sodium channel blocker mexiletine. EEG recordings were carried out in a home cage in awake animals, and the average signal power and coherence between pairs of electrodes in the range between 3 and 30 Hz were used as quantitative signal characteristics. To correlate the analgesic activity of the drugs with their EEG effects, a thermal hyperalgesia test was used, which assessed the thermal sensitivity of the hind paw by measuring the latency of the withdrawal reflex in response to temperature exposure. All three models were characterized by an increase in average signal power in the contralateral somatosensory and prefrontal cortex. Changes in coherence were found only in the neuropathic pain model (PFC and S1) and only late in development, 14 days after ligation. According to the authors, this phenomenon may be a reflection of the process of neuroplasticity that accompanies the transition of pain from acute to chronic. Ibuprofen, although active in the thermal hyperalgesia test, had no effect on the EEG spectral power, which is likely due to its ability to block only induced pain but not spontaneous pain [6]. Pregabalin and mexiletine returned the S1-PFC power and coherence values in models of inflammatory and neuropathic pain to those of healthy animals, which correlated with the results of the thermal hyperalgesia test.

#### 4.4.3. Evaluation of Antidepressant Effects

When modelling experimental depression in rodents, pronounced quantitative changes in the amplitude-spectral characteristics of EEG occur [132,133,134]. In this regard, it is possible to assess the antidepressant effect of drugs based on their effect on the parameters of the bioelectrical activity of the brain. Kudelina et al., using a model of chronic unpredictable stress in rats simulating depressive disorder, assessed the effects of the melatonin MT_1_ and MT_2_ agonist and antagonist of the serotonergic 5-HT_2c_ receptor agomelatine [7]. EEG recording was carried out on days 7, 14, and 21 of valdoxan administration. Electrodes were implanted bilaterally into the area of the somatosensory cortex and the CA1 zone of the hippocampus, and, after three days, the background EEG was recorded. The RSP (%) of the δ-, θ-, α-, and β-frequency ranges of the EEG were analyzed. Against the background of experimental depression in rats, there was a pronounced decrease in the RSP of the θ-rhythm and a twofold increase in the activity of the δ-rhythm. The administration of the drug contributed to the normalization of the bioelectrical activity of the brain in animals, namely, a significant increase in the activity of the θ-rhythm, which dominated in healthy animals.

#### 4.4.4. Evaluation of Antipsychotic Activity

Pharmaco-EEG as a tool for quantitatively assessing the effectiveness of pharmacotherapy in schizophrenia is supported by the results of a very recently published systematic review by De Pieri et al. [13]. The analysis included 22 studies out of the 1232 initially selected, in which the relationship between the clinical effects of the first-, second-, and third-generation antipsychotic drugs and EEG changes during treatment were used as endpoints. The included studies assessed patients’ power spectrum at rest and during task performance, connectivity, microstate analysis [135], and signs of epiactivity. Before treatment, the predictors of an unfavorable outcome were changes in the power of the θ-rhythm compared to the control, the high coherence and power of the α-rhythm, and a decrease in β-activity. EEG signs of effective treatment were an increase in the power of the θ- and α-rhythms, a decrease in β-activity, and a decrease in the coherence of the θ-, α-, and β-rhythms. While recognizing the need for further research in this area, the authors, nevertheless, conclude that EEG is a promising method for establishing a predictive biomarker of the brain response to drugs [13].

The work of Kalitin et al. [8] showed the effectiveness of the antipsychotic clozapine and 5-HT_2A_-antagonist RU-31 in the model of schizophrenia in rats. To model the pathology, the aspiration destruction of the ventral hippocampus was performed on day 7 of postnatal development. On day 51 of postnatal development, ECoG electrodes were implanted epidurally in the projection area of the ventral hippocampus symmetrically on both sides. It was found that, in animals with hippocampal destruction, there is a statistically significant increase in power in the α-band and β-band, as well as a decrease in power in the δ-band compared to conditionally healthy animals (only with implanted electrodes). The 5-HT_2A_ antagonist RU-31 in the group of animals with schizophrenia promoted an increase in indicators in the δ-frequency range and a decrease in the α-band relative to animals without treatment. Clozapine increased the power of the EEG signal in animals with the neonatal destruction of the ventral hippocampus in all frequency ranges studied. The results obtained correlated with the results of the Open Field test, in which animals with the destruction of the hippocampus had pronounced vertical and horizontal motor hyperactivity. The administration of RU-31 and clozapine brought the number of rearings and crossed squares to the values of animals without damage. Thus, the studied substances contributed to the correction of behavioral disorders manifested in hyperactivity, as well as electrophysiological changes caused by the surgical procedure, while similar effects were practically not observed in healthy animals.

## 5. Discussion

Based on the analysis of recent publications in which the authors use pharmaco-EEG, we can conclude that the method makes it possible to quantitatively assess the effect of drugs on the bioelectrical activity of the brain in rats under various pathologies and functional states. Currently, all kinds of commercial and homemade electrodes are available, that allow the recording of both the activity of the cerebral cortex epidurally and subdurally, and, if necessary, the underlying structures. Most researchers evaluate the amplitude-spectral characteristics of the ECoG signal calculating the powers of the δ-, θ-, α-, β-, and γ-rhythms and the degree of the inter- and intrahemispheric functional connections (“connectivity”) using cross-correlation, coherence, and some other more modern methods of analysis. The analysis of the effect of drugs on the functioning of the sensory systems can be performed by recording visual, somatosensory, or auditory evoked potentials.

The undoubted primacy in the number of publications belongs to EEG studies of antiepileptic drugs. Because of the desire of researchers to standardize such experiments, guidelines have been published that address many key issues beyond the scope of the study of epilepsy. Due to the absence of any significant alternatives, these guidelines can be used not only by researchers studying antiepileptic drugs, but, in general, by all research groups using this method in their work.

Quantitative pharmaco-EEG is a sensitive method for assessing the neuroprotective, hypnotic, analgesic, antidepressant, and antipsychotic effects of drugs, as evidenced by the publications cited in this review. Despite the popularity and many advantages of behavioral and molecular genetic methods, pharmaco-EEG certainly deserves attention as a tool that has its own unique advantages and can occupy an important niche in biomedical research.

The results of the work of Dimpfel et al. and Krijzer et al., as well our own studies, indicate that the pharmaco-EEG method in laboratory animals can be used to classify and predict the effects of drugs that affect the functions of the central nervous system. The encouraging achievements of Dimpfel et al. prove that this experimental approach not only has not remained a rudiment of neurobiological research but also can be successfully used to solve research problems nowadays. Based on the results of our own research and comparing them with the achievements of other authors, we can conclude that existing machine-learning methods can be used to solve the problems of classifying and predicting the effects of psychoactive drugs based on their EEG effects.

Regardless of the form in which the “ideal” classification algorithm will be developed (standalone software, script for MATLAB, etc.) when creating the library, the authors will have to answer a number of important questions (Figure 4). First of all, it is necessary to clearly define the type and material of the electrodes used, and their number and location. Simple electrodes like screw electrodes, on the one hand, are cheap and do not require a high level of experimental surgery during implantation. On the other hand, more expensive microelectrode arrays will have a high spatial resolution and will allow us to record more phenomena of the drug action on the brain bioelectrical activity in rodents. Most importantly, the electrode configuration chosen once to collect a library of drug effects “binds” researchers to it for many years, since a comparison of signals recorded from several thick screw electrodes with those of a microelectrode array, even if possible, generates new calculation errors. Other important issues are the signal recording parameters, the sampling rate, and the use of filters. Wideband recording (from 0.5 to 100 Hz or more) will provide more information about the action of drugs, as confirmed by studies on the efficacy of antiepileptic drugs [69,70]. Before the recording of drugs effects on EEG/ECoG, it is necessary to select the most reference agents, which can be ones already used in clinical practice and/or selective ligands (such as the NMDA-antagonist MK-801, the dopamine D2 receptor blocker raclopride, etc.). After signal registration, in order to obtain the quantitative characteristics of drug effects on EEG/ECoG, we need to answer the questions: What methods of analyzing the bioelectrical activity of the signal should be chosen? How many parameters will be sufficient for the correct work of the selected classifier? An analysis of the data from the bioelectrical activity of the brain makes it possible to calculate tens and even hundreds of amplitude-spectral characteristics of the signals, but, for the successful use of various classifiers, adequate approaches to reducing the dimensionality or identifying indicators that are most sensitive to the pharmacological effects are necessary. It is especially important that, with the emergence of new methods of machine learning, the data obtained do not lose their relevance and can provide fertile ground for research work not only for neuropharmacologists, but also for specialists in the fields of mathematics and information technology.

## 6. Conclusions

In general, the analysis of published works shows that the ECoG method can significantly enhance the objectivity of the data from pharmacological studies in rodents, thereby increasing their translational potential. The ECoG method is less susceptible to the subjectivity that can be present in behavioral and functional tests. Unlike most molecular, genetic, biochemical, and histologic methods of research, with the help of ECoG, it is possible to evaluate the condition of experimental animals repeatedly in dynamics, which corresponds to the generally accepted 3R concept. The main limitation of the method that prevents its widespread use is the need for the fabrication and surgical implantation of electrodes, which may cause difficulties for some labs. Nevertheless, a number of methodology and review articles have now been proposed that offer possible solutions. The development and widespread use of new methods of neurophysiological data analysis, as well as machine-learning methods, suggest that ECoG may become a serious tool for researchers in various fields of neuropharmacology in the foreseeable future.

## Figures and Tables

**Figure 1 brainsci-14-00772-f001:**
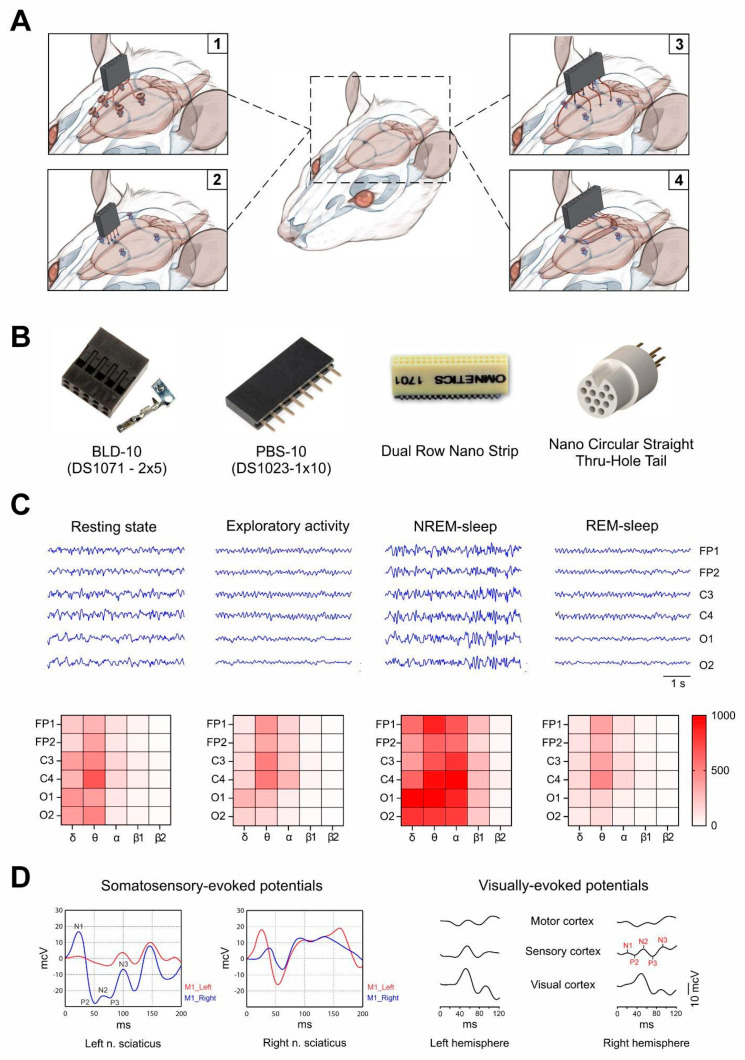
(**A**) The main types of ECoG electrodes used in experiments on rodent: 1—screw electrodes, 2—needle electrodes, 3—electrodes made of wire, 4—electrode array. (**B**) Examples of sockets used to combine electrodes into one headplug. (**C**) ECoG signal in rats at different functional states of the nervous system. Below are heat maps of average power (μV^2^) of δ- (“delta”, 1–4 Hz), θ- (“theta”, 4–8 Hz), α- (“alpha”, 8–12 Hz), and β- (“beta”, 12–30 Hz) rhythms; FP1/FP2, C3/C4, and O1/O2 are frontal, parietal, and occipital cortical areas, respectively. (**D**) Examples of somatosensory (SSEPs) and visual evoked potentials (VEPs). To elicit SSEPs, the left and right sciatic nerve were alternately stimulated (current intensity—2 mA, rectangular wave, stimulus duration—0.1 ms, and frequency—1 Hz). VEP was induced by photostimulation (stimulation frequency was 3 Hz, stimulation duration—30 s, and stimulus duration—50 ms). When analyzing evoked potentials (EPs), it is customary to distinguish negative (N) and positive (P) peaks and calculate their latencies, amplitudes, and interpeak intervals (in ms and μV). Examples of ECoG signal, SSEP, and VEP are taken from a previously published paper [20].

**Figure 2 brainsci-14-00772-f002:**
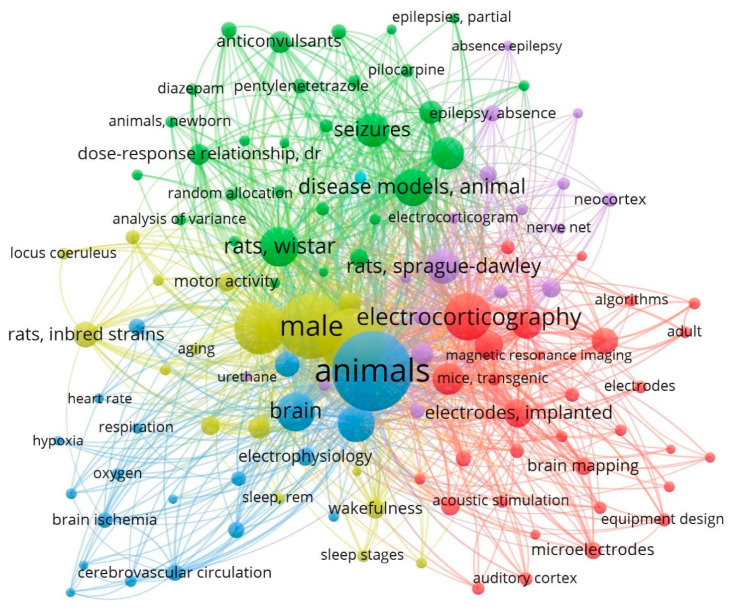
Generated term map (network visualization) based on bibliographic data of 1002 articles. Keywords from articles were parsed, analyzed, and visualized by VOSviewer. Bibliographic database file was generated based on the query “ECoG in rodents” in the PubMed system. Of the 3413 keywords, there were 119 terms that appeared in 15 and or more articles and, hence, were included in the term map. Each bubble represents a keyword; the bubble size indicates its occurrence. If two terms co-occurred more frequently, the two bubbles would be in closer proximity. Bubble color indicates that the keyword belongs to one of the 6 selected clusters. As can be seen from this term map, the algorithm identifies a whole cluster (green) related to Pharmaco-ECoG studies of epilepsy.

**Figure 3 brainsci-14-00772-f003:**
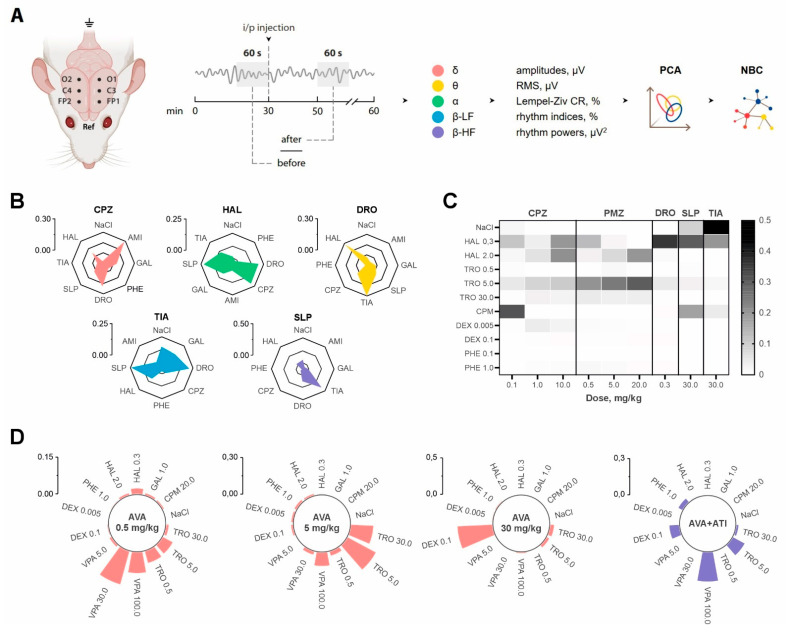
(**A**) Stages of the experiments described in [9,104,105]: I/p—intraperitoneal; RMS—root mean square; CR—compression ratio; LF—low-frequency; HF—high-frequency; PCA—principal component analysis; NBC—naïve Bayes classifier; Ref—reference electrode. (**B**) Radar charts of mean match probabilities for the pharmacological effects of chlorpromazine (CPZ), haloperidol (HAL), droperidol (DRO), tiapride (TIA), and sulpiride (SLP), and those of the drugs from the training set, as predicted by NBC in [105]. AMI—amitriptyline, GAL—galantamine, PHE—phenazepam. (**C**) A heatmap for the median identity probability values for the pharmacological effects of chlorpromazine (CPZ, 0.1, 1, and 10 mg/kg), promethazine (PMZ, 0.5, 5, and 20 mg/kg), droperidol (DR)), sulpiride (SLP), and tiapride (TIA) and the training set drugs (vertical axis), obtained with the use of NBC in [104]. HAL—haloperidol, TRO—tropicamide, CPM—chloropyramine, DEX—dexmedetomidine, PHE—phenazepam. (**D**) Radar bar plots for the median identity probability values for the valproic acid aminoester (AVA) pharmacological activity at the doses of 0.5, 5, and 30 mg/kg, and AVA with the α2-adrenergic blocker atipamezole (ATI) with the training set drugs, obtained using NBC in [9]. CPM—chloropyramine; GAL—galantamine; HAL—haloperidol; PHE—phenazepam; DEX—dexmedetomidine; VPA—sodium valproate; TRO—tropicamide.

**Figure 4 brainsci-14-00772-f004:**
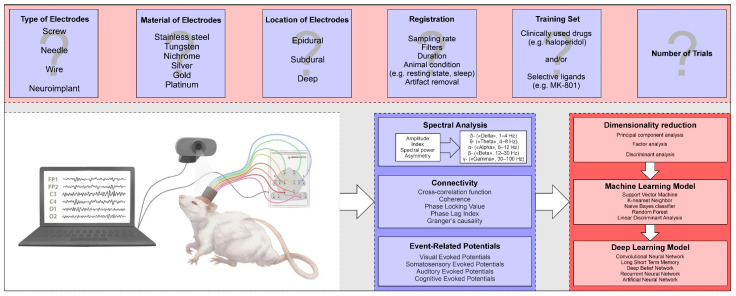
Overview of data collection, feature extraction, and classification approaches for drug activity prediction based on EEG signals. Inspired by and adapted from Sharma et al. [136].

**Table 2 brainsci-14-00772-t002:** Methodological standards for EEG studies in rodents prepared by AES/ILAE.

Publication	Questions Discussed	Reference
Methodological standards and interpretation of video-electroencephalography in adult control rodents. A TASK1-WG1 report of the AES/ILAE Translational Task Force of the ILAE.	Anesthesia, antibiotics, and analgesic used for electrode placement surgeries Types of EEG recordings and electrodes Electrode placement: considerations on surgical technique Amplifiers and filters Montages Electrical, electrode, muscle, and movement artifacts Control EEG traces in rodents Factors influencing control EEG patterns	[71]
Methodologic recommendations and possible interpretations of video-EEG recordings in immature rodents used as experimental controls: A TASK1-WG2 report of the ILAE/AES Joint Translational Task Force.	Anesthetic agents Electrode types and placement Recording issues Wired or tethered recordings Wireless recordings Artifacts Features of the EEG in immature rodents and patterns of background activity	[28]
Methodological standards and functional correlates of depth in vivo electrophysiological recordings in control rodents. A TASK1-WG3 report of the AES/ILAE Translational Task Force of the ILAE	Types of electrodes Data acquisition Types of recordings Tools for analysis Interpreting the data: LFPs and single units in information processing and cognition Relevance to human recordings	[39]
Standards for data acquisition and software-based analysis of in vivo electroencephalography recordings from animals. A TASK1-WG5 report of the AES/ILAE Translational Task Force of the ILAE. Epilepsia.	Data acquisition Signal conditioning Signal processing and analysis Data storage and data sharing	[72]

**Table 3 brainsci-14-00772-t003:** The main EEG/ECoG phenomena observed in rats after a traumatic brain injury, stroke, in neurometabolic disorders and neurodegenerative processes.

Model	Phenomena	Time	References
**Traumatic brain injury**			
*Controlled cortical impact injury (sensorimotor cortex area and underlying structures)*	↑ mean amplitude and index of δ-rhythm; ↓ mean amplitude and index of θ-, α-, and β-rhythms.	Days 3 and 7 after the injury	[32]
	↓ CCR of interhemispheric and intrahemispheric connections, as well as average coherence powers of δ-, θ-, α-, and β-rhythms.		[31]
	↑ latency of N1 and N3 peaks of VEP in the area of trauma on day 3 after the trauma with subsequent normalization by day 7; ↑ amplitude of P2 peak on day 3; ↓ P2 peak amplitude on day 7.		[33]
	Decreased amplitudes of early (N1 and P2) and late (N2, P3, and N3) SSEP components, increased latency of early and shortened latency of late waves.	Day 7 after the injury	[76]
**Ischemic stroke**			
*Bilateral ligation of the common carotid arteries*	↑ RSP of δ-rhythm and ↓ RSP of θ-, α-, and β-rhythms in somatosensory cortex.	24 h after the ligation	[78]
*Temporary occlusion of MCA (30 and 45 min)*	↑ index of δ-rhythm and ↓ index of θ-, α-, and β-rhythms in the ischemic hemisphere on day 3 after 45 min MCA occlusion; ↑ mean amplitude, index, and mean power of θ-rhythm on day 7; ↓ CCR of interhemispheric and intrahemispheric connections.	Days 3 and 7 after the occlusion	[80]
	↓ amplitude of P2 and N3 peaks and P3–N3 interpeak interval of ipsilateral SSEPs; ↓ amplitude of P3 peak and duration of N2–P3 interpeak interval of contralateral SSEPs.	Day 7 after the occlusion	[79]
*Temporary occlusion of MCA (90 min)*	↑ power of δ-rhythm in sensorimotor cortex in the acute phase; ↓ power of α-rhythm in sensorimotor cortex in the subacute phase.	Acute phase (3 h and 6 h); subacute phase (12, 24, 48, and 72 h); and chronic period (96, 120, 144, and 168 h).	[77]
**Neurometabolic disorders**			
*Hepatic encephalopathy caused by hepatic artery ligation or administration of 1000 mg/kg galactosamine*	“Leftward shift” of the spectral power.	4–5 h (hepatic artery ligation); and 30 h (administration of galactosamine)	[81]
*Diabetes: BB/Wor rats*	↑ VEP latency.	Month 6 and 12	[82,83]
*Non-alcoholic fatty liver disease: 35% fructose solution for 8 weeks*	↑ the number of spikes after pentylenetetrazole administration.	8 weeks	[84]
**Neurodegenerative diseases**			
*A model of parkinsonism induced by intranigral administration of the neurotoxin 1-methyl-4-phenylpyridinium ion (MFP+)*	Registration of synchronous discharges of high-amplitude slow waves in the range of δ- and θ-frequencies with amplitude of 150–200 μV in SMC, SN, and CN (24 h); ↑ δ-activity in CN (24 h); ↑ θ-activity in SMC and SN (24 h); ↑ power of θ-rhythms in SMC, SN, and CN (day 7); ↑ β-activity in the SMC of the right hemisphere (day 7).	24 h and day 7	[85]

Note: RSP—relative spectral power; SMC—sensorimotor cortex; MCA—middle cerebral artery; SSEP—somatosensory evoked potentials; VEP—visual evoked potentials; CCR—cross-correlation coefficient; SN—substantia nigra; CN—caudate nucleus.

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
