# Peer review of "Prospects of Electrocorticography in Neuropharmacological Studies in Small Laboratory Animals"

_brainsci, 2024, doi:10.3390/brainsci14080772_

Round 1

Reviewer 1 Report

Comments and Suggestions for Authors

Overall, this is a comprehensive review/overview of the topic. However, some parts are a bit superficial and do not go into significant depth that is needed. This includes the subsections of 'BASIC METHODS OF ANALYZING BIOELECTRICAL ACTIVITY OF THE BRAIN'. Additionally, the article could really benefit from having more visualizations/figures. Finally, Figure 2 is not very informative. The texts in the figure are barely visible and the figure it not sufficiently well integrated into the text.

Author Response

Comment 1: Overall, this is a comprehensive review/overview of the topic. However, some parts are a bit superficial and do not go into significant depth that is needed. This includes the subsections of 'BASIC METHODS OF ANALYZING BIOELECTRICAL ACTIVITY OF THE BRAIN'.

Response 1: We understand that the section 'BASIC METHODS OF ANALYZING BIOELECTRICAL ACTIVITY OF THE BRAIN' is written rather superficially, but in this review, we do not focus on a detailed description of the mathematical foundations of EEG signal analysis. Excellent works have been published on this topic, for example, Im, C.H. Computational EEG Analysis; Im, C.-H., Ed, Springer Singapore: Singapore, 2018 and Chiarion, G.; Sparacino, L.; Antonacci, Y.; Faes, L.; Mesin, L. Connectivity Analysis in EEG Data: A Tutorial Review of the State of the Art and Emerging Trends. Bioengineering 2023, 10, 372, which we refer to in our work. The purpose of this review is to introduce the reader to possible areas of application of pharmaco-EEG/ECoG and to show that the method can be a useful addition to behavioral and molecular genetic research methods. In general, this article is addressed more to “classical” pharmacologists or neuroscientists involved in the study of the pharmacological activity of new molecules, rather than to engineers or mathematicians.

Comment 2: Additionally, the article could really benefit from having more visualizations/figures.

Response 2: We agree with the reviewer’s assessment. We have added more figures to the manuscript.

Comment 3: Finally, Figure 2 is not very informative. The texts in the figure are barely visible and the figure it not sufficiently well integrated into the text.

Respone 3:We kindly appreciate the reviewer’s feedback, but we would like to propose to change portrait orientation of fig.2 to landscape one. Also fig. 2 has been transferred to discussion section with additional description in the text.

Reviewer 2 Report

Comments and Suggestions for Authors

The current manuscript is an interesting study on electrocorticography applied in neuropharmacological studies in small laboratory animals. It is well written, structured, and easy to read, with some representative images. Thus, I only advise on the following alterations before acceptance for publication:

- Other similar reviews have been published, such as the links that are given bellow; thus (although they already do so, but only superficially), the authors should better support the novelty of their review, and mention what it is that this manuscript brings to this specific scientific field if accepted for publication;

https://www.ncbi.nlm.nih.gov/pmc/articles/PMC4706962/

https://karger.com/nps/article/72/3-4/139/233796/Pharmaco-EEG-Studies-in-Animals-A-History-Based

https://onlinelibrary.wiley.com/doi/10.1155/2016/8213878

- All titles should begin in caps, the manuscript’s title should be corrected;

- The manuscript’s formatting is not in accordance with the journal’s guidelines, this should be corrected;

- In the introduction section, before mentioning electrocorticography, other relevant neurophysiological methods should be mentioned, and these should be compared with the selected topic of the review;

- In the introduction section, more should be said on the disadvantages of behavioral tests, real-time PCR, western blotting, enzyme immunoassay, etc., when compared to EEG and ECoG;

- More representative images should be produced and added, namely in what concerns: the advantages of electrocorticography when compared to other (still more used) methods; the different types of electrodes, present in table 1; the different mentioned methods for analyzing brain bioelectrical activity; evaluation of hypnotic effects, analgesic activity, antidepressant effects and antipsychotic activity;

- The conclusion section is too long, this section should really be a “final discussion” section, and a short and summarizing conclusion section should be added afterwards;

- An abbreviation list is missing and should be added.

Author Response

Comment 1: The current manuscript is an interesting study on electrocorticography applied in neuropharmacological studies in small laboratory animals. It is well written, structured, and easy to read, with some representative images.

Response 1:Thank you!

Comment 2:Thus, I only advise on the following alterations before acceptance for publication:

- Other similar reviews have been published, such as the links that are given bellow; thus (although they already do so, but only superficially), the authors should better support the novelty of their review, and mention what it is that this manuscript brings to this specific scientific field if accepted for publication;

https://www.ncbi.nlm.nih.gov/pmc/articles/PMC4706962/

https://karger.com/nps/article/72/3-4/139/233796/Pharmaco-EEG-Studies-in-Animals-A-History-Based

https://onlinelibrary.wiley.com/doi/10.1155/2016/8213878

Response 2: As suggested by the reviewer, additional information has been added in the introduction section

Comment 3: All titles should begin in caps, the manuscript’s title should be corrected;

Response 3: Manuscript’s title has been corrected

Comment 4: The manuscript’s formatting is not in accordance with the journal’s guidelines, this should be corrected;

Response 4: Main text and reference style formatting has been prepared in accordance with the journal’s guidelines

Comment 5: In the introduction section, before mentioning electrocorticography, other relevant neurophysiological methods should be mentioned, and these should be compared with the selected topic of the review;

Response 5: Additional information has been added in the introduction section

Comment 6: In the introduction section, more should be said on the disadvantages of behavioral tests, real-time PCR, western blotting, enzyme immunoassay, etc., when compared to EEG and ECoG;

Response 6: Thank you for pointing this out. Disadvantages of behavioral tests, real-time PCR, western blotting, enzyme immunoassay have been added to the text

Comment 7: More representative images should be produced and added, namely in what concerns: the advantages of electrocorticography when compared to other (still more used) methods; the different types of electrodes, present in table 1; the different mentioned methods for analyzing brain bioelectrical activity; evaluation of hypnotic effects, analgesic activity, antidepressant effects and antipsychotic activity;

Response 7: Thank you for pointing this out. Unfortunately, we are restrained in short deadline, therefore we are unable to make several new pictures in time. But we added two representative images have been added to the manuscript: network visualization of ECoG research in rodents (figure 2) and use of ECoG in drug activity classification and prediction (figure 3). We kindly hope that these figures will increase the quality of our review.

Comment 8: The conclusion section is too long, this section should really be a “final discussion” section, and a short and summarizing conclusion section should be added afterwards;

Response 8: Thank you for this suggestion. Manuscript’s structure has been changed in accordance with your advice.

Comment 9: An abbreviation list is missing and should be added.

Response 9: Unfortunately, abbreviation list is not provided by the journal format. We define abbreviations in each of three sections: the abstract; the main text; the first figure or table. When defined for the first time, abbreviations were added in parentheses after the written-out form. We did not find any abbreviation list in previously published MDPI reviews. If Brain Sciences Editorial Office makes it possible to add abbreviation list, we will add it in the manuscript. 

Reviewer 3 Report

Comments and Suggestions for Authors

The review article "Prospects of electrocorticography in neuropharmacological studies in small laboratory animals" describes the various types of electrodes and their application in neurological studies. However, certain points need to be revised and focused. These include,

1. The author needs to clearly describe the various types of electrodes instead of giving information distributed elsewhere. They need to remove Table 1 and write the electrodes in small para with the advantages and disadvantages of those electrodes in research.

2. Unnecessary wording such as "Thanks to this property" in line 154 needs to be removed.

3. The use of EEG in various neurological studies even in clinical settings is well known and is being used in epilepsy and insomnia, thus the author needs to concise the detail they described and make it short.

4. Similarly, the "evaluation of neuroprotective activity" should be shortened and focused, with the change in heading the use of ECoG in the evaluation of neuroprotective activity. 

5. Need to add a para describing how electrocardiography with other biomarkers will enhance the outcome of the preclinical and clinical study. The author also needs to add why various preclinical or clinical studies are unable to add electrocardiography for neurological disease. One factor could be the inability of the instruments in their facility due to lack of funding, therefore these labs depend only on certain biomarkers.  

Comments on the Quality of English Language

NA

Author Response

The review article "Prospects of electrocorticography in neuropharmacological studies in small laboratory animals" describes the various types of electrodes and their application in neurological studies. However, certain points need to be revised and focused. These include,

Comment 1: The author needs to clearly describe the various types of electrodes instead of giving information distributed elsewhere. They need to remove Table 1 and write the electrodes in small para with the advantages and disadvantages of those electrodes in research.

Response 1: Thank you for this suggestion. We would like to keep Table 1 in the manuscript but we added more information about advantages and disadvantages of those electrodes in the text.

Comment 2: Unnecessary wording such as "Thanks to this property" in line 154 needs to be removed.

Response 2: Thank you for pointing this out. Unnecessary wording in line 154 has been removed.

Comment 3: The use of EEG in various neurological studies even in clinical settings is well known and is being used in epilepsy and insomnia, thus the author needs to concise the detail they described and make it short.

Comment 4: Similarly, the "evaluation of neuroprotective activity" should be shortened and focused, with the change in heading the use of ECoG in the evaluation of neuroprotective activity. 

Response 3 and 4: We kindly appreciate the reviewer’s feedback, but we respectfully disagree with suggestions 3 и 4. In this review, we did not focus on the use of the method in the diagnosis of epilepsy and insomnia, but on examples of the use of pharmaco-EEG in rodents to study the effects of antiepileptic and neuroprotective agents. We have given some examples of the use of Pharmaco-EEG in clinical practice to show that such method is not the field of experimental biology alone, but can be effectively used at the level of clinical research.

Comment 5: Need to add a para describing how electrocardiography with other biomarkers will enhance the outcome of the preclinical and clinical study. The author also needs to add why various preclinical or clinical studies are unable to add electrocardiography for neurological disease. One factor could be the inability of the instruments in their facility due to lack of funding, therefore these labs depend only on certain biomarkers.  

Response 5: Thank you for pointing this out. Additional information has been added in «Electrodes for recording of brain bioelectrical activity» and «conclusion» sections.

Round 2

Reviewer 3 Report

Comments and Suggestions for Authors

The authors answered my concerns